# Direct measurement of nanostructural change during in situ deformation of a bulk metallic glass

Thomas C. Pekin[1,2], Jun Ding [3], Christoph Gammer [4], Burak Ozdol[2], Colin Ophus[2], Mark Asta[1,3], Robert O. Ritchie [1,3] & Andrew M. Minor [1,2]

To date, there has not yet been a direct observation of the initiation and propagation of individual defects in metallic glasses during deformation at the nanoscale. Here, we show through a combination of in situ nanobeam electron diffraction and large-scale molecular dynamics simulations that we can directly observe changes to the local short to medium range atomic ordering during the formation of a shear band. We observe experimentally a spatially resolved reduction of order prior to shear banding due to increased strain. We compare this to molecular dynamics simulations, in which a similar reduction in local order is seen, and caused by shear transformation zone activation, providing direct experimental evidence for this proposed nucleation mechanism for shear bands in amorphous solids. Our observation serves as a link between the atomistic molecular dynamics simulation and the bulk mechanical properties, providing insight into how one could increase ductility in glassy materials.

[1] Department of Materials Science and Engineering, University of California, Berkeley, CA 94720, USA. [2] National Center for Electron Microscopy, Molecular Foundry, Lawrence Berkeley National Laboratory, 1 Cyclotron Road, Berkeley, CA 94720, USA. [3] Materials Sciences Division, Lawrence Berkeley National Laboratory, 1 Cyclotron Road, Berkeley, CA 94720, USA. [4] Erich Schmid Institute of Materials Science, Austrian Academy of Sciences, Jahnstrasse 12, 8700 Leoben, Austria. Correspondence and requests for materials should be addressed to T.C.P. (email: tcpekin@berkeley.edu) or to A.M.M. (email: aminor@lbl.gov)

Bulk metallic glasses (BMGs) are an interesting class of materials noted for their wide variety of mechanical properties, associated most notably with their lack of long-range crystallographic order[1]. BMGs include alloys that exhibit extremely high strength in excess of most engineering materials[2,3], as well as low stiffness and high elastic strain limits[4,5]. Because of these wide-ranging properties, BMGs are attractive alloys for future applications as they offer a potential for the development of stronger and tougher structural materials[6–9].

One of the main impedances to the adoption of high-strength BMGs is their limited ductility, which can be restricted by single shear band formation and rapid propagation at low strains, which often results in catastrophic failure[10–14]. Accordingly, of key importance to further alloy development is understanding how such shear bands originate at the nanoscale because, although single shear-band formation can cause BMGs to fail at near-zero tensile ductilities, multiple shear-band formation represents the fundamental essence of plasticity in these alloys. As BMGs invariably display high strength, the creation of tensile ductility — via multiple shear banding — is thus essential to their fracture toughness, and hence damage-tolerance, in terms of their potential role as future structural materials.

Several mechanisms have been proposed for the initiation and propagation for shear bands, the predominant hypotheses being free volume softening[15–20], adiabatic heating softening[21–23], and shear transformation zones (STZs)[20,24,25]. Recently, with advancements in both modeling[26–28] and experimentation[29], STZ formation has emerged as the prevailing mechanism by which shear bands form and propagate[10,11]. In this theory, a STZ is a cluster of atoms which plastically rearranges under mechanical stress. As the stress to transform many STZs homogeneously is very high, in a real material this is hypothesized to preferentially occur at stress concentrations[10,30]. Once a high enough density of activated STZs have formed, a shear band develops and can propagate[10,11].

To date, observing the mechanisms of BMG shear band formation, while possible in molecular dynamics (MD) simulations, has been experimentally challenging due to the high rate of the catastrophic shear band propagation and the current experimental limits of electron microscopy. However, observing shear band nucleation and dynamics at the scales possible in transmission electron microscopy (TEM) is crucial to linking our understanding of deformation mechanisms provided by MD simulations to the macroscale mechanical behavior. Previous TEM experiments in bulk metallic glasses have largely been limited to ex situ qualitative imaging studies with high enough resolution to resolve shear bands, but have difficulties in quantitative interpretation[31–34], or more quantitative fluctuation electron microscopy (FEM) studies on the structure of BMGs[35–39] that fall below the local spatial resolution needed for individual shear band characterization. In situ experiments to date have been qualitative, too slow during acquisition, hard to interpret due to a lack of understanding of the contrast mechanisms in shear bands, or at too low of a spatial resolution to be comparable to MD models[40]. Recent advancements in techniques and hardware have, however, allowed for the observation of strain[41] and as we will show here, the evolution of locally resolved atomic short and medium range order, with nanometer resolution during in situ deformation, providing much more comparable information to the significant modeling efforts which have been performed.

In this study, we design an in situ sample to study the coupling of local atomic order and strain during tensile deformation. The BMG used in this study is a member of the model glass family $Cu_xZr_yAl_{100-(x+y)}$[42,43], which has been extensively studied for its high glass-forming ability[44], and relative ease of computational modeling. These glasses have local clusters of atoms that pack into icosahedral structures[45–49], which due to their two-, three- and five-fold symmetry axes in projection, have characteristic symmetric diffraction patterns[50]. We directly observe a change in structural order correlated with strain as measured from the NBED patterns acquired during in situ deformation.

## Results

**In situ nanobeam electron diffraction.** Specifically, the sample used was $Cu_{46}Zr_{46}Al_8$, which was thinned to electron transparency (~80–90 nm) and then milled using a focused ion beam (FIB) into an in situ tensile bar specimen. The annular dark field images (ADF) of the resulting sample and subsequent deformation can be seen in Fig. 1a. Unique to this in situ experiment, after each 10 nm increase in deformation, deformation was paused and a nanobeam electron diffraction (NBED) dataset was acquired, in which a full diffraction pattern was acquired for each ADF image pixel at 400 frames per second, for a total of 167,440 diffraction patterns. The full experimental procedures can be found in the attached methods section. The diffraction patterns were then used to measure the spatially resolved evolution of both strain[41] and short and medium range order[48–52] at every probe position over a large area as the sample was mechanically deformed, with a spatial resolution (probe position step size) of 2.5 nm. It should be noted that these diffraction patterns arise through an interaction of the electron beam with a finite sized (1.47 nm FWHM) volume projected through the sample thickness, and therefore symmetry elements in the patterns can arise from interactions with multiple oriented clusters, making it impossible in this experiment to distinguish between singular oriented clusters (short range order) and cluster networks (medium range order). In addition, it has been shown that as the sample thickness increases, higher order symmetries are extinguished in the diffraction patterns[51].

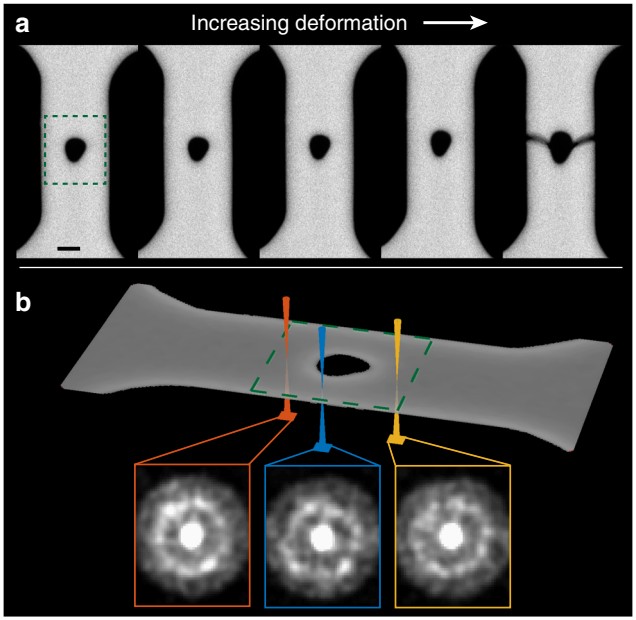

**Fig. 1** A schematic overview of the experiment. **a** Annular dark field (ADF) scans taken before, during, and after in situ deformation of a bulk metallic glass sample. The change in the size and shape of the hole is indicative of plastic deformation. The lack of contrast change across the specimen suggests constant sample thickness. The green dashed box shows where NBED was performed. The scale bar is 150 nm. **b** Schematic showing the NBED process in the metallic glass. As the beam rasters over the area, a full map consisting of over 33,000 nanobeam electron diffraction patterns is recorded. The three patterns shown are examples containing two-fold (orange), four-fold (blue), and zero (yellow) symmetries, respectively

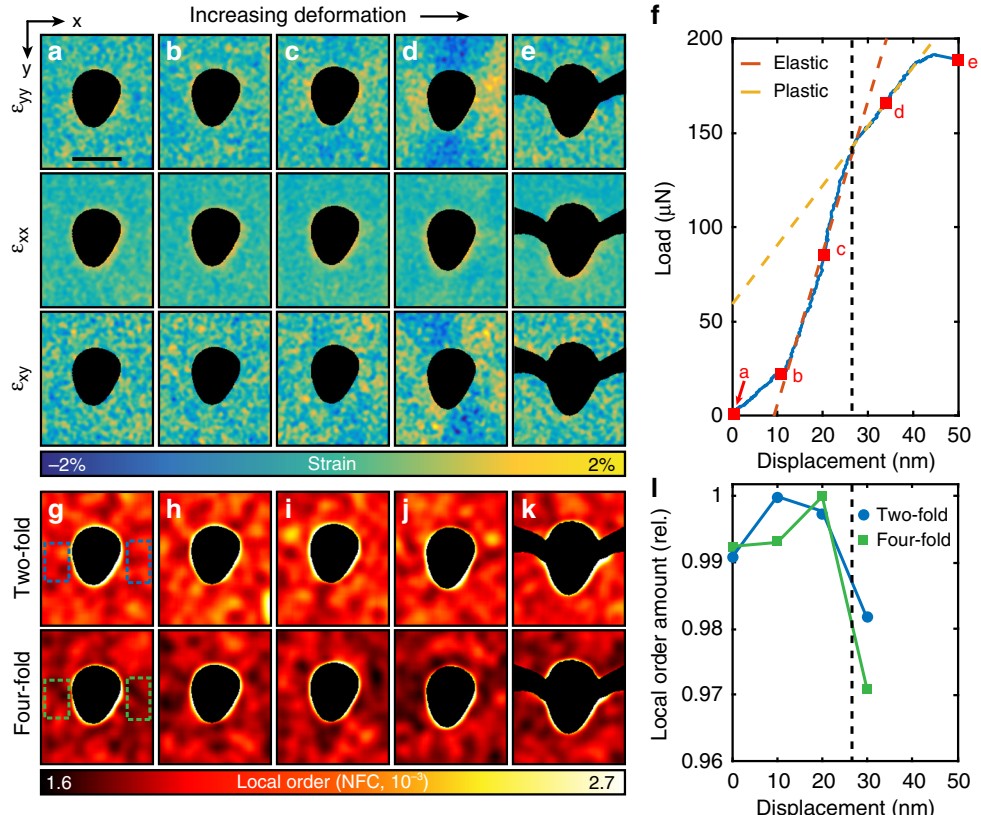

**Fig. 2** The results of strain and order mapping during in situ deformation. **a–e** Strain maps at increasing deformation steps. The scale bar is 150 nm. The top row is strain in the tensile direction, the middle row is strain perpendicular to the tensile direction, and the bottom row of strain maps is shear. By step **d** plastic strain has developed in the tensile direction. **f** Load-displacement plot acquired from the in situ TEM mechanical testing holder. The blue solid line is the data acquired, with the displacements corresponding to the different map acquisitions **a–e** marked with red squares. The orange dashed line is the least squares fit to the elastic regime, while the yellow dashed line corresponds to the plastic region. The change in slope between **c**, **d** is indicative of plastic deformation. The sample fractured before **e**. The drift in the plot corresponding to the hold times during the NBED acquisitions at the 10, 20, and 30 nm steps has been removed. **g–k** Maps showing local order at increasing deformation steps as measured from the diffraction patterns. The top row corresponds to two-fold symmetry, while the bottom row corresponds to four-fold symmetry. The scale corresponds to the normalized Fourier coefficients (NFC). **l** A plot showing the relative mean amount of two-fold (blue circle) and four-fold (green square) order in the corresponding areas in **g** at successive deformations. These areas correspond to 3450 diffraction patterns. The plot shows roughly a 2–3% reduction order relative to the maximum over the entire area

Figure 2 shows the results obtained after processing the NBED data acquired during the in situ experiment. The top three rows of images (Fig. 2a–e) show relative strain as the experiment progresses in the orthogonal, and shear directions, while the bottom two rows of images (Fig. 2g–k) show the degree of two and four-fold order. As the experimental conditions were optimized for strain mapping, we were unable to resolve significant amounts of other rotational symmetries in the diffraction patterns. This was likely due to thickness effects. A clear evolution of strain is observed during deformation. The strain concentrates itself as expected around the hole in the center of the sample, as well as along the shear directions 45° to loading directions. Examining the maps more closely, during the first three steps of deformation corresponding to 0, 10, and 20 nm of displacement, very little changes with regards to strain. These steps correspond to elastic loading of the sample, which can be observed in the load-displacement curve shown in Fig. 2f. As the sample is deformed to 30 nm, nonlinearity of the load-displacement curve begins to occur, indicative of plastic strain. In the strain maps at this deformation (Fig. 2a–e), clear strain concentrations occur on both sides of the FIB milled hole. These thin areas experience strain up to 2% above the median strain of the sample, leading to local regions of plastic deformation and

failure. After fracture, the strain returns to a uniform value across the field of view.

In conjunction with the evolution of local strain, we also measured a spatially resolved reduction in diffraction pattern symmetry (indicative of a change in the local atomic clustering) with deformation. The results can be seen in Fig. 2g–k. As previously shown in the literature[50], by measuring the symmetry elements found in each diffraction pattern, we can spatially map order. As expected, initially the sample had a uniform distribution of order across the region of interest. This is expected for a rapidly quenched bulk metallic glass. This does not measurably change during the elastic deformation of the sample (10 and 20 nm of deformation). However, once plasticity begins, we observe the destruction of atomic short and medium range order spatially confined to the regions of high plastic strain. In the two-fold order map Fig. 2j, this destruction is confined to the highest strain region on the sample, on the right side of the hole. In the four-fold order map (Fig. 2j, bottom), this destruction is more prevalent and can be seen on both sides of the hole, but again, is confined to the high strain regions of the sample. Figure 2l shows the mean of the Fourier symmetry coefficients in the color-coded rectangles shown in Fig. 2g for the first four steps before fracture. These sub-regions comprise of 3450 patterns. Within these

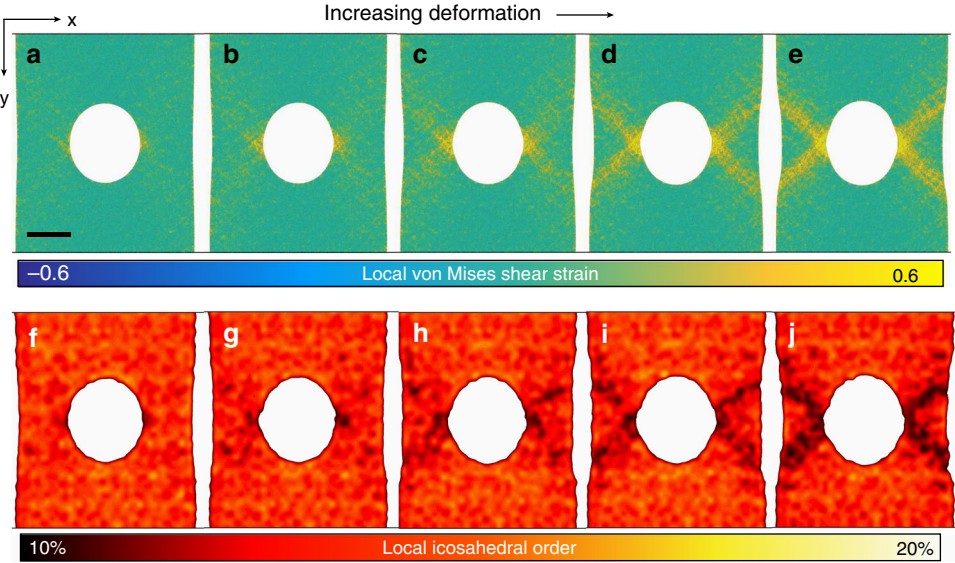

**Fig. 3** Five frames of MD-simulated $Cu_{46}Zr_{46}Al_8$ MG at the strain of 4, 5, 6, 7, and 8% respectively, under uniaxial tensile deformation. The scale bar is 20 nm. **a–e** A color map of local von Mises shear strain. **f–j** The spatial distribution of local icosahedral order in coarse-graining scale, for such five frames

plastically deforming regions, there is a ~2–3% reduction in order from the maximum for both two- and four-fold symmetries. A rigorous statistical analysis confirming the reduction in local order is given in the Supplementary Note 1, as well as a second choice of measurement area in Supplementary Fig. 1. Furthermore, Supplementary Fig. 2 plots the NFCs versus strain over the whole field of view (>22,000 probe positions) before and during deformation. While there is no correlation between strain state and amount of order in the unstrained state, there exists a small but measurable negative correlation between strain and order under deformation.

The difference in size between the two- and four-fold symmetry areas of reduced order is most likely due to the mechanics of order destruction and diffraction, namely that the clusters of projected four-fold symmetry have more avenues to move out of Friedel diffraction than those in two-fold symmetry. Additionally, it should be noted that since this measurement is projected through the sample thickness, if four-fold symmetry arises from the overlap of two two-fold symmetry elements, this arrangement of atoms is more easily displaced as to reduce diffraction during deformation than a single cluster, further contributing to an increased likelihood of reduction of four-fold order when compared to two-fold order. It must be emphasized that, due to plural scattering and projection effects inherent to this experiment, decrease in any symmetry order coefficient (while indicative of structural change) is difficult to directly map to equivalent reductions of symmetry in individual singular clusters of atoms.

**Molecular dynamics simulations**. To help interpret these results, large-scale molecular dynamics (MD) simulations[53] were performed on a 5-million-atom sample with a similar geometry, although with reduced dimensions and higher strain rates for practical considerations. The alloy composition was the same as tested experimentally. The parity between experimental and simulated sample geometry and composition allows for their correlation. Details of sample preparation can be found in the supplied Methods. The MD results can be seen in Fig. 3 that shows five frames of increasing deformation. The process of strain localization around the region of plastic deformation is shown in Fig. 3a–e; the images are colored according to the local von Mises shear strain (see Methods). Under the uniaxial tension in the *y*-direction, the strain localization begins to aggregate near both sides of the hole (Fig. 3a); with increase in applied strain, progressively more strain localization evolves out along the two 45° directions of maximum shear stress. Finally, the continuously induced strain localizations percolate across the sample from the hole to the free surfaces, thus forming the shear bands seen in Fig. 3e. This deformation process revealed by the MD simulation is generally consistent with our experimental TEM characterization shown in Fig. 2. Similarly, Fig. 3f–j show the spatial distribution of local icosahedral order (see Methods) for the studied five frames, where a corresponding reduction in the fraction of full icosahedral order (as a proxy for local rotational symmetry) is seen as deformation progresses. Interestingly, the regions with reduced local icosahedral order (blue regions in Fig. 3f–j) nearly overlap with the strain localization in Fig. 3a–e. Comparing our large-scale MD simulation results and our direct observations by NBED, a clear correlation between strain localization and local structural transition in BMGs under deformation can be seen in both datasets.

Despite experimental differences in sample size and deformation rate, previous and current MD simulations observe a mechanism of metallic glass deformation which relies on STZs collective activation and plastic rearrangement as the basis for shear band formation. In simulations, the result of the STZs deformation was locally reduced icosahedral order, i.e., a destruction in the icosahedral clusters followed by subsequent shear band initiation. Similarly, while in this particular experiment it is not possible to directly link the structural transition mechanism under deformation to icosahedral order, we do see for the first time structural transition under deformation in the in situ NBED experiment. At high strains, multiple local atomic clusters containing symmetry are destroyed before shear band propagation, and their destruction generates increasing local plasticity and shear band formation under further strain. The experimental determination of the exact structural transition mechanism will require further experiments with a higher time resolution and thinner specimens, most likely utilizing the same NBED technique.

## Discussion

Our results support the important hypothesis of shear transformation zone activation leading to shear banding and fracture as described by prior computational models but never experimentally observed before. In order to improve the limited tensile ductility currently preventing BMGs from widespread use,

resulting primarily from deformation localization in single shear bands, our experiments suggest that future alloy design should follow a pathway which allows for more STZ activation to homogeneously occur before critical failure. We also believe that the combination of in situ microscopy and NBED as a characterization technique is well suited to image structural deformation characteristics on the nanometer scale in nominally disordered materials. The direct correlation between quantitative in situ deformation experiments and large-scale MD simulations on the same length scale can serve as a crucial link between simulations and bulk mechanical properties.

## Methods

**Materials.** The samples were bulk metallic glass with a composition of $Cu_{46}Zr_{46}Al_8$, prepared via suction casting from the melt.

**Sample preparation and experimental methods.** The initial sample was received as a cylindrical bar with an outer diameter of ~4 mm. The bar was mechanically machined down to an outer diameter of 3 mm, and electrical discharge machined into 600-µm-thick slices. The samples were further mechanically thinned to ~150–200 µm, and then jet polished to electron transparency. The jet polishing solution used was 33% nitric acid in methanol, cooled to −25 C, and a polishing voltage of 18 V.

In situ samples were then cut and lifted out using a FEI Strata 235 dual beam focused ion beam (FIB) equipped with an Omniprobe. The samples were welded on to a Hysitron push-to-pull chip using deposited platinum. A hole was cut in the center of the tensile bar to concentrate stress and strain, allowing for higher resolution scans in the critical area. The sample after liftout can be seen in Supplementary Fig. 3.

The in situ nanobeam electron diffraction (NBED), pre- and post-experiment imaging was performed on the TEAM I transmission electron microscope[54], a $C_s$ and $C_c$ corrected FEI microscope. The microscope was operated in scanning transmission electron microscopy (STEM) mode with an accelerating voltage of 300 kV. The microscope was operated in three condenser lens nanoprobe mode, with a spot size of 10, 10 µm aperture, and a convergence angle of 0.91 mrad. The probe was measured in real space to have a full width half maximum diameter of 1.47 nm.

In situ deformation was performed in the TEM using a Hysitron PI-95 in situ deformation holder, with the sample mounted to a push to pull device. The sample was deformed under displacement control, in steps of 10 nm, up to a total displacement of 40 nm when the sample broke. NBED datasets with a size of 182 by 184 probe positions were acquired with a Gatan K2-IS camera at 400 frames per second, with a probe step size of 2.5 nm. Each scan took 70 s while deformation was paused. During this time there was negligible spatial drift of the sample due to the high stability of both the microscope and holder, which was confirmed by the lack of drift artifacts in the simultaneously acquired HAADF images. The camera acquired a 1792 by 1920 pixel full-frame diffraction pattern at each real space pixel location. Five complete datasets were acquired, corresponding to tensile deformations of 0, 10, 20, 30, and 40 nm. The sample broke at some point between 30 and 40 nm of deformation. The load-displacement curve acquired during active deformation can be seen in Fig. 2f. The load has been plotted after a removal of linear load drift with time, which reduces vertical jumps in the plot during NBED acquisition. The load-displacement does not show pure elasticity, instead at high deformations, plasticity is observed. The unadulterated load and displacement curves with respect to time can be seen in Supplementary Fig. 4.

The NBED datasets were acquired in a sparse electron scattering regime, which was controlled by reducing the beam current to the sample. This allowed for the counting of individual electron locations, resulting in a massive reduction in the amount of data, and a large increase of signal to noise. Once the patterns were reduced to electron locations, patterns with reduced noise were reconstructed. These patterns were shifted to remove the effect of beam sway during scanning. For the strain mapping analysis, every pattern was locally summed with its nearest neighbors, using a Gaussian weighting centered on the center pixel with a standard deviation width of two pixels. These diffraction patterns were then used for strain mapping following the methods outlined in ref. [41], in which an ellipse is fitted to every diffraction pattern using Eq. 1, and deviations from a reference radius are converted to strains. This is done using the following equations, $\epsilon_{xx} \approx 1/2(A - 1)$, $\epsilon_{xy} \approx 1/2B$, and $\epsilon_{yy} \approx 1/2(C - 1)$, from ref. [41]. The reference radius was chosen to minimize strain at zero loading.

The mapping of local symmetry followed the methods proposed in refs. [50,51], after correction for elliptical astigmatism produced by both microscope aberrations as well as strain. To determine the ellipticity, we have fitted the ring intensity $I(q_x, q_y)$ over the reciprocal space coordinates $(q_x, q_y)$ using the following form

$$I(q) = I_r \exp\left[-\frac{\left(q_0 - \sqrt{Aq_x^2 + Bq_xq_y + Cq_y^2}\right)^2}{2s^2}\right] + I_0 + I_1\sqrt{q_x^2 + q_y^2}, \tag{1}$$

where $I_r$, $I_0$, and $I_1$ are the intensities of the ring, constant background, and linear background, respectively. $A$, $B$, and $C$ are the ellipse coefficients, and $s$ is the standard deviation of the ring, equivalent to its width. $(q_x, q_y)$ is centered on the center of the diffraction pattern. The resulting fitted ellipse can then be represented in matrix form as

$$\mathbf{M}_{\text{ellipse}} = \begin{bmatrix} A & B/2 \\ B/2 & C \end{bmatrix} \tag{2}$$

To remove elliptical distortions, each electron location $(q_x, q_y)$ can be transformed by

$$(q_x', q_y') = (q_x, q_y) \times \mathbf{v} \times \sqrt{\mathbf{d}} \times \mathbf{v^T} \tag{3}$$

where $\mathbf{v}$ is the eigenvectors of $\mathbf{M}_{\text{ellipse}}$, $\mathbf{d}$ is the eigenvalues corresponding to $\mathbf{v}$, and $\mathbf{v^T}$ is the transpose of $\mathbf{v}$.

The symmetry of each pattern then is measured from the binned polar transforms of the electron coordinates for each pattern to $(k, \varphi)$ space, as explained in refs. [48,50,51]. Similar to the cited references, to compute the symmetries in the pattern, we used the four-point angular cross correlation function (CCF), as a function of scattering vector. This takes the form

$$\text{CCF}(\mathbf{r}, \mathbf{k}, \Delta) = \frac{\langle I(k, \varphi)I(k, \varphi + \Delta) \rangle_\varphi - \langle I(k, \varphi) \rangle_\varphi^2}{\langle I(k, \varphi) \rangle_\varphi^2}, \tag{4}$$

where $I(k, \varphi)$ is the intensity diffracted into the $(k, \varphi)$ bin, $\langle \rangle_\varphi$ is the average with respect to $\varphi$ at a fixed $k$, and $\mathbf{r}$ is the STEM probe position, $(x, y)$. Then, by taking the absolute value of the Fourier transform of these CCFs, the symmetries in the diffraction pattern can be accessed directly from the appropriate (already normalized) Fourier coefficient. This method removes the effects of changing diffraction pattern intensity (for example from sample thickness, bending, etc.). A histogram of the two and four-fold symmetry coefficients is provided in Supplementary Fig. 5.

**Computational molecular dynamics methods.** Large-scale molecular dynamics simulations were implemented to study the $Cu_{46}Zr_{46}Al_8$ model metallic glasses, using the optimized embedded atom method (EAM) potential, adopted from ref. [55]. The sample contained about 5 million atoms, and the liquids of the sample were equilibrated for 1 ns at high temperature (2500 K) to assure equilibrium and then quenched to room temperature at the cooling rates of $10^{12}$ K s$^{-1}$ employing a Nose-Hoover thermostat (the external pressure was barostated at zero)[53]. Periodic boundary conditions were applied in all three directions during the quenching[53]. The prepared $Cu_{46}Zr_{46}Al_8$ metallic glass sample had dimensions of 78.3 nm × 91.5 nm × 13.1 nm at 300 K. A hole with the radius of 16 nm was created in the middle of $x - y$ plane. Then the boundary condition in $x$-direction was set as a free surface. The as-processed sample was then gradually heated to 680 K (below its glass transition temperature $T_g$) and annealed for 0.5 ns to reach a steady potential energy. The final step of sample preparation was to quench it from 680 to 300 K with a cooling rate of $10^{12}$ K s$^{-1}$. For the simulation of the deformation process, the $Cu_{46}Zr_{46}Al_8$ model metallic glass was under uniaxial tension in the $y$-direction with a strain rate of $10^{-4}$ ps$^{-1}$ at 300 K under NVT ensemble. The local von Mises shear strain was analyzed using the algorithm in ref. [56], by comparing the deformed configuration with the original one. We monitored the local structural order in $Cu_{46}Zr_{46}Al_8$ metallic glasses by conducting Voronoi tessellation[57]. Faces of Voronoi cell with area smaller than 0.25% of the total area were discounted. Specifically, Fig. 3f–j are the coarse-graining plots of the fraction of full icosahedral order (with the Voronoi index $\langle 0, 0, 12, 0 \rangle$) in the $x$–$y$ plane, which is divided into pixels with dimensions of $2 \times 2$ nm and averaged over the whole $z$-direction of the sample. The stress–strain curve and corresponding reduction in local order are shown in Supplementary Fig. 6.

## Data availability

Data is available via request to the corresponding authors.

## Code availability

All code is available via request to the corresponding authors.

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

## Acknowledgements

We would like to acknowledge Linzhi Zhao, Yanhui Liu, and Weihua Wang for providing the as-processed BMG samples. This work was funded by the U.S. Department of Energy, Office of Science, Office of Basic Energy Sciences, Materials Sciences, and Engineering Division under Contract No. DE-AC02-05-CH11231 within the Mechanical Behavior of Materials program (KC13). The electron microscopy work at the Molecular Foundry was supported by the Office of Science, Office of Basic Energy Sciences, of the U.S. Department of Energy under contract no. DE-AC02-05CH11231. This research used resources of the National Energy Research Scientific Computing Center (NERSC), a U.S. Department of Energy Office of Science User Facility operated under contract no. DE-AC02-05-CH11231.

## Author contributions

M.A., R.O.R., and A.M.M. conceived the project. T.C.P, C.G., and B.O. performed the sample preparation and NBED experiment. T.C.P. and C.O. performed the analysis on the experimental data, including software development. J.D. developed and performed the MD simulation and analysis. T.C.P., with help from J.D., wrote the original draft, which was reviewed and edited by all authors. Project administration, supervision, and funding acquisition was performed by M.A., R.O.R., and A.M.M.

## Additional information

**Competing interests:** The authors declare no competing interests.

