## [Peer Review File · Nature Communications]

Reviewers' comments:

Reviewer #1 (Remarks to the Author):

The plastic deformation mechanism of metallic glasses has been the topic for many researches. Although the variation of local atomic order during the deformation process has been investigated in many atomistic simulations, a direct experimental observation/confirmation is however not available yet. The present work reported a direct experimental observation of the deformation process of a metallic glass, finding a reduction in local atomic order upon plastic deformation of the metallic glass. The experimental observation agree well with the simulation results, not only revealing the atomic process of the shear band formation, but also proving the reliability of atomistic simulations in the study of metallic glasses. In this sense, although the finding is not earthshaking, it is of great importance for the community in deepening the understanding of the plastic deformation of metallic glasses. I would therefore like to recommend its publication in NCOMM.

Albeit, I have two comments/suggestions for the authors to consider:

- 1) The observed reduction in 2-fold and/or 4-fold order is said to be "reduction of short and medium range order", how does one distinguish the short range order and the medium range order in metallic glasses? The observed local icosahedral order in simulations is apparently of short range (nearest neighbors).
- 2) The load-displacement and local order-displacement curves are shown in Fig. 2f and 2i. Similar curves are expected in Fig. 3 for a quantitative comparison between experiments and simulations.

Reviewer #2 (Remarks to the Author):

This manuscript reports pioneering results on the measurement of elastic strain and structural order in a metallic glass under deformation. State-of-the-art transmission electron microscopy, with nanobeam electron diffraction, allows mapping of strain and order with a resolution of some 2 nm. This mapping is correlated with the results of molecular-dynamics atomistic simulations. Although the simulations involve much shorter length and time scales than the physical experiments, and the simulated glass is likely to be much less relaxed than the real glass, it is nevertheless possible to make worthwhile comparisons.

Although the results are novel, the new insights gained are limited – essentially confirmation that plastic deformation begins with structural disordering even ahead of shear-band formation.

In the Materials and Methods section, it should be made clear that 'Fig. 1' and 'Fig. 2' are referring to Supplementary Figures.

Reviewer #3 (Remarks to the Author):

The paper reports the measurement of the change in the local structure using electron nano diffraction in an in situ tensile loading setting. The in situ methodology used in this work is novel. While the experimental method is still limited in some aspects (e.g. frame rate), I see high potential in further developing the method to acquire more sophisticated information in the near future.

Despite the novel approach, the manuscript has numerous issues, mainly the lack of depth in understanding the current issues in the field. This has led to numerous overstatements on what new information is provided from the new experiment described in this paper, and how it may relate to the information provided by the MD simulation. There are also some lack of details on the

data analysis. All of these issues are critical, and therefore I cannot recommend this paper to be published in Nat Comm.

The very first sentence in the abstract says “ there has never been a direct observation of the initiation and propagation of individual defects in metallic glass during deformation”. This is an overstatement. Earlier efforts to achieve this have involved several other in situ microscopic techniques, including optical microscopy and infrared measurements (e.g. see Yang et al, *Intermetallics* 12, 1265 (2004)).

The sentence in the abstract “This is the first direct experimental confirmation of this proposed nucleation mechanism for shear bands in amorphous solids”, is also an overstatement, for the reasons that will be explained below.

MD simulation is used to simulate shear bands. These simulations, while they provide useful insights, cannot be accepted as fully realistic simulations, as they use empirical potentials developed primarily based on other atomistic simulations (e.g. DFT) or experimental data (e.g. PDF) that have their own limitations. For example, it has never been clear if these models (both DFT and EAM) can generate realistic medium range order (MRO) structure at the nanoscale. This means that the potential MRO-shearband interaction in the deformation simulation could be misleading. The current paper uses the MD simulation results almost as the reference to their experimental data, which can lead to misguided information. Deformation simulations, either atomistic or mesoscale, can be used to guide our understanding, but they do have their own significant limitations and therefore great care is required when drawing any conclusions from the experiment-simulation comparison.

In page 3, it says “Previous TEM experiments in bulk metallic glasses have largely been limited to ex situ qualitative imaging studies, or more quantitative fluctuation electron microscopy (FEM) studies on the structure of BMGs that fall below the local spatial resolution needed for individual shear band characterization”. This is not true, as TEM techniques have always had high enough spatial resolution to observe shearbands which have appeared to be at least a few nanometers thick. What has limited in those experiments were rather the temporal resolution, which is still the problem, even in the work reported in this paper. One other thing to note is that, although there have been many TEM works that showed what appear to be shaerbands, understanding the contrast mechanism of those shearband-like features still remains very difficult. More specifically, there has been lack of understanding on why the shaerbands appear to have less density in those images, as such low density cannot always be resulted according to the current deformation theories (e.g. STZ theory).

In page 3, it says “These glasses have clusters of atoms that pack into stable icosahedral structures”, but the sentence needs to be stated in a more subtle way. It is undeniable that the icosahedral clusters are energetically stable, but there is still ongoing debate on whether the atomistic models overly stabilize those clusters due to the fast quenching rate in the simulation. These models always tend to generate significant amount of icosahedral clusters (although the amount varies depending on the composition), and there has been no quantitative (experimental) information that can verify such simulation results (None of the references in this paper provides that information).

It may be worth indicating what exactly the “frame rate” is in this experiment. How many ADF images can be generated per second? That’s an important number because eventually it is what sets the temporal limits of the in situ measurement. The paper shows some relevant numbers, such as 33,000, 167,440 patterns or etc, but it is difficult to properly convert those numbers into the exact frame rate. And how does the frame rate compare to the deformation rate, or any known shear band propagation rate?

What is the estimated thickness of the TEM specimen (along the beam direction)? This is important

because the specimen thickness can greatly impact the rotational symmetry that appears in the diffraction patterns. It may also potentially involve the breakdown of Friedel symmetry (this is discussed in Liu et al., the Ref. 48). Plural scattering can also substantially affect the data.

In page 5, the sentence "It must be noted that these diffraction patterns arise through an interaction of the electron beam projected through the sample thickness, and therefore symmetry elements in the patterns can arise from interactions with multiple oriented clusters, making it impossible in this experiment to distinguish between short and medium range order" needs to be further clarified. If the authors intended to use something similar to Liu's (Ref. 48) or Hirata's (Ref. 49) method, then it may have been challenging already because the probe size was ~ 2nm, which I believe is much bigger than the size used in those references. What does it exactly mean by "distinguish between short and medium range order"? What length scale distinguishes between SRO and MRO?

It is not clear how the rotational symmetry (2, 3, or 4 fold) was determined from those patterns. The authors might have used similar methods reported in those references, but I believe the exact calculation method needs to be described in this paper, as it is important. There is only marginal description on this in the supplementary text.

In page 5, it says "we were unable to observe other rotational symmetries in the diffraction patterns". The statement needs to be more quantitative. For example, was there absolutely no 5-fold symmetry at all? What about 6, 7, 8, or 10-folds? Those tend to appear even due to some random correlations within the illuminated volume (e.g. Ref. 48).

In page 7, it says "In conjunction with the evolution of local strain, we also measured a spatially resolved change in short and medium range order, or the atomic clustering of the metallic glass with deformation". There is an issue with this statement. What the authors have measured is the change in the rotational symmetry appeared within those patterns, but it does not provide the direct information on how the symmetries correlate to SRO, MRO, or any atomic clustering. In other words, there is a gap between what has been measured, and what the authors think they measured. For the same reason, "destruction of atomic short and medium range order" mentioned below needs to be revised as well. There is no direct link between the disappearance of rotational symmetry and the destruction of SRO or MRO, as the authors do not know what those SRO or MRO is, although it can be speculated that there may be a potential link.

Same page, "As previously shown in the literature (Ref 48), by measuring the symmetry elements found in each diffraction pattern, we can spatially map order". This needs to be revised in a more careful way. The previous works, including Ref 48, have tried to detect (or map) spatial ordering (or heterogeneity) in similar but different ways. Their results may vary, and there is still ongoing debate on which information is reliable, and what conclusion can be drawn from the data.

Same page, "there is a ~7% reduction in order from the maximum for both two- and four-fold symmetries", but it is impossible to get any quantitative sense on what the 7% means, as the scale bar in the figure (Figure 2 g to k) do not have any unit. This relates to the point I made earlier on how the symmetries were calculated. How did you get the "normalized 4-fold order coefficient" shown in Supplementary Fig. 3?

In page 9, I believe the sentence "It is therefore apparent that the results of our large-scale MD simulation are fully consistent with the direct observation by nanobeam electron diffraction microscopy that correlates the strain localization and local structural transition in BMGs under deformation" is an overstatement, based on the reasoning that I provided on the limitations of the simulation above.

Same page, "In simulations, the result of the STZs deformation was a more disordered region as icosahedral clusters are destroyed, and subsequent shear band initiation". I disagree with this

statement, as it is a typical misguided statement made only from the perspective of MD simulation where icosahedral clusters are the “ordering” and the rest is disordered. Unfortunately, this particular view point has misled the community for a long time. There can be other types ordering that may potentially form in MG structure (especially at the MRO scale) without involving any icosahedra (see Tanaka et al, Nat Mater 9, 324 (2010)).

Reviewers' comments:

Reviewer #1 (Remarks to the Author):

The plastic deformation mechanism of metallic glasses has been the topic for many researches. Although the variation of local atomic order during the deformation process has been investigated in many atomistic simulations, a direct experimental observation/confirmation is however not available yet. The present work reported a direct experimental observation of the deformation process of a metallic glass, finding a reduction in local atomic order upon plastic deformation of the metallic glass. The experimental observation agree well with the simulation results, not only revealing the atomic process of the shear band formation, but also proving the reliability of atomistic simulations in the study of metallic glasses. In this sense, although the finding is not earthshaking, it is of great importance for the community in deepening the understanding of the plastic deformation of metallic glasses. I would therefore like to recommend its publication in NCOMM.

Albeit, I have two comments/suggestions for the authors to consider:

1) The observed reduction in 2-fold and/or 4-fold order is said to be "reduction of short and medium range order", how does one distinguish the short range order and the medium range order in metallic glasses? The observed local icosahedral order in simulations is apparently of short range (nearest neighbors).

The reviewer brings up a good point with regards to the differences between short and medium range order. We have clarified a sentence on page 5 in which short and medium range order is better defined, as well as clarified the abstract a bit by removing an instance of "short to medium range" when discussing our observations. Reviewers 1 and 3 are correct in that there is absolutely a distinction between short and medium range orders, however, in this experiment we can only measure the contributions of both simultaneously, as experimentally it is difficult to say if the structured diffraction is from only a cluster, or a network of clusters projected through the sample thickness. This is discussed more in the main body of the paper now as well.

2) The load-displacement and local order-displacement curves are shown in Fig. 2f and 2i. Similar curves are expected in Fig. 3 for a quantitative comparison between experiments and simulations.

We have added a Supplementary Figure 3 that contains the data requested. Thank you for the suggestion.

Reviewer #2 (Remarks to the Author):

This manuscript reports pioneering results on the measurement of elastic strain and structural order in a metallic glass under deformation. State-of-the-art transmission

electron microscopy, with nanobeam electron diffraction, allows mapping of strain and order with a resolution of some 2 nm. This mapping is correlated with the results of molecular-dynamics atomistic simulations. Although the simulations involve much shorter length and time scales than the physical experiments, and the simulated glass is likely to be much less relaxed than the real glass, it is nevertheless possible to make worthwhile comparisons.

Although the results are novel, the new insights gained are limited – essentially confirmation that plastic deformation begins with structural disordering even ahead of shear-band formation.

In the Materials and Methods section, it should be made clear that ‘Fig. 1’ and ‘Fig. 2’ are referring to Supplementary Figures.

Thank you for pointing this out, we have clarified in the Materials and Methods to which figure we are referencing.

Reviewer #3 (Remarks to the Author):

The paper reports the measurement of the change in the local structure using electron nano diffraction in an in situ tensile loading setting. The in situ methodology used in this work is novel. While the experimental method is still limited in some aspects (e.g. frame rate), I see high potential in further developing the method to acquire more sophisticated information in the near future.

Despite the novel approach, the manuscript has numerous issues, mainly the lack of depth in understanding the current issues in the field. This has led to numerous overstatements on what new information is provided from the new experiment described in this paper, and how it may relate to the information provided by the MD simulation. There are also some lack of details on the data analysis. All of these issues are critical, and therefore I cannot recommend this paper to be published in Nat Comm.

The very first sentence in the abstract says “ there has never been a direct observation of the initiation and propagation of individual defects in metallic glass during deformation”. This is an overstatement. Earlier efforts to achieve this have involved several other in situ microscopic techniques, including optical microscopy and infrared measurements (e.g. see Yang et al, Intermetallics 12, 1265 (2004)).

The reviewer is correct in that other people have watched shear band formation and propagation in other microscopy configurations. However, this is the first to our knowledge at this size scale and the first utilizing NBED. We have clarified that this is the first experiment in which we observe structural transitions “at the nanoscale”.

The sentence in the abstract “This is the first direct experimental confirmation of this proposed nucleation mechanism for shear bands in amorphous solids”, is also an overstatement, for the reasons that will be explained below.

In conjunction with the comments below, we have changed ‘confirmation’ to ‘evidence’, as this more accurately describes the data we collected.

MD simulation is used to simulate shear bands. These simulations, while they provide useful insights, cannot be accepted as fully realistic simulations, as they use empirical potentials developed primarily based on other atomistic simulations (e.g. DFT) or experimental data (e.g. PDF) that have their own limitations. For example, it has never been clear if these models (both DFT and EAM) can generate realistic medium range order (MRO) structure at the nanoscale. This means that the potential MRO-shearband interaction in the deformation simulation could be misleading. The current paper uses the MD simulation results almost as the reference to their experimental data, which can lead to misguided information. Deformation simulations, either atomistic or mesoscale, can be used to guide our understanding, but they do have their own significant limitations and therefore great care is required when drawing any conclusions from the experiment-simulation comparison.

In addition to changing ‘confirmation’ to ‘evidence’, we have also restructured how we describe the MD comparisons in the abstract. Specifically, we state that the reduction of order in the simulations is due to the hypothesized mechanism in the real experiment, shear transformation zone activation. We have expanded on the nuances and differences between reduction of diffraction pattern symmetry and local order more in the body text. We believe that in combination with the change to experimental evidence, rather than confirmation, of the proposed mechanism seen in the MD simulation, that the statement no longer is an overstatement.

In page 3, it says “Previous TEM experiments in bulk metallic glasses have largely been limited to ex situ qualitative imaging studies, or more quantitative fluctuation electron microscopy (FEM) studies on the structure of BMGs that fall below the local spatial resolution needed for individual shear band characterization”. This is not true, as TEM techniques have always had high enough spatial resolution to observe shearbands which have appeared to be at least a few nanometers thick. What has limited in those experiments were rather the temporal resolution, which is still the problem, even in the work reported in this paper. One other thing to note is that, although there have been many TEM works that showed what appear to be shaerbands, understanding the contrast mechanism of those shearband-like features still remains very difficult. More specifically, there has been lack of understanding on why the shaerbands appear to have less density in those images, as such low density cannot always be resulted according to the current deformation theories (e.g. STZ theory).

We have reconfigured this sentence to reflect that the reviewer is correct in that TEM was always able to resolve shear bands in imaging mode, with difficulties in interpretation, while contrasting this with quantitatively interpretable STEM FEM, which by design lacks a true spatial resolution (while containing multitudes of structural information). We additionally mentioned these caveats in the *in situ* experiments to date, which only complicate interpretation further, and motivate our study.

In page 3, it says “These glasses have clusters of atoms that pack into stable icosahedral structures”, but the sentence needs to be stated in a more subtle way. It is undeniable that the icosahedral clusters are energetically stable, but there is still ongoing debate on whether the atomistic models overly stabilize those clusters due to the fast quenching rate in the simulation. These models always tend to generate significant amount of icosahedral clusters (although the amount varies depending on the composition), and there has been no quantitative (experimental) information that can verify such simulation results (None of the references in this paper provides that information).

We have removed the word stable such that the sentence reads “pack into icosahedral structures”, and would like to mention that references 46, 47, 50, and 51 all contain experimental evidence behind icosahedra in CuZr BMGs.

It may be worth indicating what exactly the “frame rate” is in this experiment. How many ADF images can be generated per second? That’s an important number because eventually it is what sets the temporal limits of the *in situ* measurement. The paper shows some relevant numbers, such as 33,000, 167,440 patterns or etc, but it is difficult to properly convert those numbers into the exact frame rate. And how does the frame rate compare to the deformation rate, or any known shear band propagation rate?

We have edited the manuscript to state that the frame rate of acquisition is 400 frames per second, as well as in the main manuscript mention that deformation was paused during image acquisition. The reviewer is correct in that the frame rate is directly tied to the ability to see shear bands propagate, it must be noted that we do not actually capture shear band propagation, but rather precursors to initiation. During the pauses in the stepped deformation, a 4D “frame” was acquired in approximately 60 seconds. However, this is limited by scan size as well as camera frame rate. Decreasing the scan to 64x64 would reduce this frame rate to 10 seconds, which can be reduced even further with faster frame rates.

What is the estimated thickness of the TEM specimen (along the beam direction)? This is important because the specimen thickness can greatly impact the rotational symmetry that appears in the diffraction patterns. It may also potentially involve the breakdown of Friedel symmetry (this is discussed in Liu et al., the Ref. 48). Plural scattering can also substantially affect the data.

We thank the reviewer for pointing this out- as this was indeed an oversight that has now been corrected. The sample thickness has been emphasized in both the main text and the methods, as well as the effects that the sample thickness has in the data acquired (namely a reduction in higher order symmetries).

In page 5, the sentence “It must be noted that these diffraction patterns arise through an interaction of the electron beam projected through the sample thickness, and therefore symmetry elements in the patterns can arise from interactions with multiple oriented clusters, making it impossible in this experiment to distinguish between short and medium range order” needs to be further clarified. If the authors intended to use something similar to Liu’s (Ref. 48) or Hirata’s (Ref. 49) method, then it may have been challenging already because the probe size was $\sim 2\text{nm}$, which I believe is much bigger than the size used in those references. What does it exactly mean by “distinguish between short and medium range order”? What length scale distinguishes between SRO and MRO?

Reviewer 1 also brought up this important point. As mentioned further below, we are using the same method as Liu, et al. Through the emphasis of the thickness in the main text and the addition of the probe size, we hope to increase the clarity behind the statement that due to the constraints with projecting through such a thickness, we cannot sample individual clusters. Instead we are sampling many. However, we cannot determine if these clusters are networked (medium range order), or randomly individually (short range order) constructively interfering along the beam direction. This is a limitation of the experimental requirements for *in situ* deformation. The question of short range order and medium range order length scale is clarified in the manuscript.

It is not clear how the rotational symmetry (2, 3, or 4 fold) was determined from those patterns. The authors might have used similar methods reported in those references, but I believe the exact calculation method needs to be described in this paper, as it is important. There is only marginal description on this in the supplementary text.

We have clarified in the Methods exactly how the symmetries are computed, which is exactly the same as Liu’s method.

In page 5, it says “we were unable to observe other rotational symmetries in the diffraction patterns”. The statement needs to be more quantitative. For example, was there absolutely no 5-fold symmetry at all? What about 6, 7, 8, or 10-folds? Those tend to appear even due to some random correlations within the illuminated volume (e.g. Ref. 48).

We have added a sentence saying that the higher order symmetries are unobservable most likely due to thickness effects.

In page 7, it says “In conjunction with the evolution of local strain, we also measured a spatially resolved change in short and medium range order, or the atomic clustering of the metallic glass with deformation”. There is an issue with this statement. What the authors have measured is the change in the rotational symmetry appeared within those patterns, but it does not provide the direct information on how the symmetries correlate to SRO, MRO, or any atomic clustering. In other words, there is a gap between what has been measured, and what the authors think they measured. For the same reason, "destruction of atomic short and medium range order" mentioned below needs to be revised as well. There is no direct link between the disappearance of rotational symmetry and the destruction of SRO or MRO, as the authors do not know what those SRO or MRO is, although it can be speculated that there may be a potential link.

This is an important point that we have now clarified. Specifically, we have adjusted this sentence to say that we measure a reduction in diffraction pattern symmetry, which is indicative of local atomic clustering, with deformation. Our reason for stating this connection between the diffraction pattern symmetry and local atomic clustering is based off of ref. 50. The reviewer is correct in that we directly measure changes in symmetry in locally resolved diffraction patterns. However, the reviewer might be missing the direct link between refs. 50 and 51 (mapping structural order based on symmetry), and our work, which employs the same technique. Additionally, refs. 48 and 49 discuss diffraction from icosahedra in BMGs. While there is no current direct link between the disappearance of rotational symmetry and the destruction of SRO or MRO under any sort of *in situ* experiment, this is part of the purpose of comparing MD simulations and experimental data in this paper. To the authors' knowledge, no one has linked the two before.

Same page, “As previously shown in the literature (Ref 48), by measuring the symmetry elements found in each diffraction pattern, we can spatially map order”. This needs to be revised in a more careful way. The previous works, including Ref 48, have tried to detect (or map) spatial ordering (or heterogeneity) in similar but different ways. Their results may vary, and there is still ongoing debate on which information is reliable, and what conclusion can be drawn from the data.

We use the same method for mapping as ref. 50 and 51 (previously was 48 before edits), which is supported by refs. 48 and 49, in which individual icosahedral diffraction patterns are observed. A similar method in ref. 52 was used to map symmetry/MRO as well.

Same page, “there is a ~7% reduction in order from the maximum for both two- and four-fold symmetries”, but it is impossible to get any quantitative sense on what the 7% means, as the scale bar in the figure (Figure 2 g to k) do not have any unit. This relates to the point I made earlier on how the symmetries were calculated. How did you get the “normalized 4-fold order coefficient” shown in Supplementary Fig. 3?

We have included more information on how these symmetries are computed in the Methods. We have changed “mean amount” to “mean of the Fourier symmetry coefficients”.

In page 9, I believe the sentence “It is therefore apparent that the results of our large-scale MD simulation are fully consistent with the direct observation by nanobeam electron diffraction microscopy that correlates the strain localization and local structural transition in BMGs under deformation” is an overstatement, based on the reasoning that I provided on the limitations of the simulation above.

We have changed this sentence to reflect the criticisms brought to our attention by the reviewer. It is now “Comparing our large-scale MD simulation results and our direct observations by NBED, a clear correlation between strain localization and local structural transition in BMGs under deformation can be seen in both datasets”.

Same page, “In simulations, the result of the STZs deformation was a more disordered region as icosahedral clusters are destroyed, and subsequent shear band initiation”. I disagree with this statement, as it is a typical misguided statement made only from the perspective of MD simulation where icosahedral clusters are the “ordering” and the rest is disordered. Unfortunately, this particular view point has misled the community for a long time. There can be other types ordering that may potentially form in MG structure (especially at the MRO scale) without involving any icosahedra (see Tanaka et al, Nat Mater 9, 324 (2010)).

We thank the reviewer for this comment. The classical theoretical model of atomic packing in metallic glasses was based on icosahedral order, which has been proposed for several decades. Such local icosahedral order in metallic glasses has been revealed by ab initio simulation (e.g. HW Sheng et al. Nature. 439, 419-425 (2006)), XRD+EXAFS+RMC (e.g. HW Sheng et al. Nature (2006)) and direct observation by nanobeam electron diffraction (e.g. A. Hirata et al. Nature Materials. 10, 28-33 (2011)). As pointed out by the reviewer, it’s true that some other works have indicated the important role of medium-range crystalline order, e.g. in the two-dimension colloidal glass (Tanaka et al Nat. Mater. 9, 324 (2010)) and three-dimension colloidal glass (Leocmach et al. Nat Commn. 3, 974 (2012)). However, we think it’s reasonable to adopt the model of icosahedral order, especially for metallic glasses, in this work, as such model is widely accepted by the metallic glass community. To clarify the results of the simulation in the text, we have modified the sentence in question to read “In simulations, the result of the STZs deformation was locally reduced icosahedral order, i.e., a destruction in the icosahedral clusters followed by subsequent shear band initiation.”

Reviewers' comments:

Reviewer #1 (Remarks to the Author):

The authors have responded to the concerns raised and revised the manuscript accordingly. I suggest to accept it for publication.

Reviewer #3 (Remarks to the Author):

The authors have added some more details on the experimental details, and also modified some texts to accommodate my previous concerns. However, the manuscript still has multiple major problems, and therefore I cannot recommend this paper for publication in Nat. Comm.

Here are the major problems:

First, there is a serious problem in the analysis in Figure 2. In the plot in Fig. 2i, the authors claim there is "destruction" of 2 and 4 fold order. But this is not convincing at all, because the plot is the result of the analysis that is only performed on the area indicated within the dashed lines. Those areas seem to be chosen manually (presumably following the regions that show the largest strain), and I believe this is a biasing in the analysis. There are plenty of other regions that show similar amount of change in the color, for example the yellow spot on the left hand side of the hole, the dark strip below the hole in Fig. 2i, or the yellow spot right hand side of the hole in Fig. 2j. Why do the authors think that these variation in colors are not as important as the change within the dashed areas? What makes the authors think the manual selection of the area (i.e. the dashed regions) can be justified for this analysis? Also, within the dashed areas, the 2-fold order initially increases before it decreases at the fourth frame, and therefore it is not a simple monotonic decrease as the authors seem to suggest. The mechanism may be more complicated according to the data.

Second, even if the claimed "destruction of 2 and 4 fold order" is real, I don't think that connects well to the reduction of the icosahedral order seen in the MD simulation. By trying to making this connection, the authors are making the error of assuming that the icosahedral clusters are the only ones that show the rotational symmetry. There are plenty of other possible structures that are not icosahedra that show 2 or 4 fold symmetry (in fact, as authors mentioned in the text, icosahedral clusters don't even intrinsically have 4 fold symmetry). The simulation data in Fig. 3 only show the reduction in the icosahedral clusters, but not other cluster types, which I think is a surprisingly naive presentation, given that the 2nd author of this manuscript (J. Ding) has already related the dynamics of the non-icosahedral clusters to the potential plasticity carrier in his earlier publication (PNAS, 2014, 111, p. 14052).

There are other issues as well:

The authors now provides the thickness information, which is 80-90nm. This is very thick. If the thickness is 80-90nm, then dynamic diffraction should dominant. And I am not sure if such condition is suitable to observe any subtle changes in the structure.

In page 5, how do you determine the spatial resolution of 2nm, when the size of the electron beam is 1.47nm FWHM?

More details of the in situ loading is provided (10nm then pause), but what about the drift rate? 70 sec per frame is very long. Any effort to correct the effect of drift?

There is still no mention about how the strain was determined. I know this was explained in Ref 41, but it will be necessary to at least have them included in the methodology section in this paper, as it is important. Reading this paper only gives no idea of how they are determined. It is

also puzzling because, if the patterns are like the ones shown in Fig.1, then I think they look quite different from the ones shown in Ref 41 (which is possibly due to the difference in the probe size?)

Reviewers' comments:

Reviewer #1 (Remarks to the Author):

The authors have responded to the concerns raised and revised the manuscript accordingly. I suggest to accept it for publication.

We would like to thank the reviewer for his helpful contributions.

Reviewer #3 (Remarks to the Author):

The authors have added some more details on the experimental details, and also modified some texts to accommodate my previous concerns. However, the manuscript still has multiple major problems, and therefore I cannot recommend this paper for publication in Nat. Comm.

Here are the major problems:

First, there is a serious problem in the analysis in Figure 2. In the plot in Fig. 2l, the authors claim there is “destruction” of 2 and 4 fold order. But this is not convincing at all, because the plot is the result of the analysis that is only performed on the area indicated within the dashed lines. Those areas seem to be chosen manually (presumably following the regions that show the largest strain), and I believe this is a biasing in the analysis. There are plenty of other regions that show similar amount of change in the color, for example the yellow spot on the left hand side of the hole, the dark strip below the hole in Fig.2i, or the yellow spot right hand side of the hole in Fig. 2j. Why do the authors think that these variation in colors are not as important as the change within the dashed areas? What makes the authors think the manual selection of the area (i.e. the dashed regions) can be justified for this analysis? Also, within the dashed areas, the 2-fold order initially increases before it decreases at the fourth frame, and therefore it is not a simple monotonic decrease as the authors seem to suggest. The mechanism may be more complicated according to the data.

Due to the nature of deformation in a metallic glass, we would expect that changes in structure would be spatially inhomogeneous, concentrated in areas of high stress, and directly related to plastic deformation. This is in fact what we see in the areas previously chosen. These choices motivated the manual selection of the area previously measured. However, as we show now explicitly, the result is consistent for areas chosen through other methodologies as well. In order to demonstrate this, we have now moved the previous analysis to the Supplementary Information, and have instead used a standardized rectangle in the high stress regions on either side of the hole. There are a few consequences of this. As deformation in metallic glasses is highly localized, we would expect to see less of decrease of order in the larger area being measured rather than the concentrated areas of large strain. This is consistent with our observations. We have additionally included more histograms of the symmetry coefficients, as well as their respective two-sample t-test results in the Supplementary Information to support

our results and demonstrate the statistical significance. We hope that this change will prove satisfactory.

With regards to the non-monotonically decrease in order, the first three steps really shouldn't have much change in order, as the sample has not yet plastically deformed. This is stated in the text. The change in the plot (2I) is the natural variation from changing sample geometry, orientation, and measurement error for the first three steps. The truly important part is the decrease after plasticity has occurred, which has now been emphasized with a dashed line for clarity in Fig. 2I.

Second, even if the claimed "destruction of 2 and 4 fold order" is real, I don't think that connects well to the reduction of the icosahedral order seen in the MD simulation. By trying to making this connection, the authors are making the error of assuming that the icosahedral clusters are the only ones that show the rotational symmetry. There are plenty of other possible structures that are not icosahedra that show 2 or 4 fold symmetry (in fact, as authors mentioned in the text, icosahedral clusters don't even intrinsically have 4 fold symmetry). The simulation data in Fig. 3 only show the reduction in the icosahedral clusters, but not other cluster types, which I think is a surprisingly naive presentation, given that the 2nd author of this manuscript (J. Ding) has already related the dynamics of the non-icosahedral clusters to the potential plasticity carrier in his earlier publication (PNAS, 2014. 111, p. 14052).

The experimental source of four-fold symmetry is explained in the text, as well as seen in refs. 48, 50 and 52, among others, and can arise from icosahedral networks, or other cluster types, and it is difficult to determine exactly which type of cluster(s) caused this diffraction effect.

After considering the reviewer's comments carefully, we are in agreement that the direct mechanistic comparison between the MD model and the experiment is not as well supported as we had previously written. Specifically, the reviewer is correct in that the MD model focuses on the icosahedral nature of order, while the experiment in its current form cannot distinguish between anything other than more or less symmetrical order. To this effect, we have updated the text stating this. While we do directly measure the structural change reflected in the diffraction patterns, as the paper's title states, we have emphasized discovering the exact structural unit responsible for the mechanism as a route for future work, which would be able to utilize the method we demonstrate in this manuscript. We thank the reviewer for leading us to this clarification.

There are other issues as well:

The authors now provides the thickness information, which is 80-90nm. This is very thick. If the thickness is 80-90nm, then dynamic diffraction should be dominant. And I am not sure if such condition is suitable to observe any subtle changes in the structure.

The thickness was mentioned prior to any revision but we have added it to the main text as it is important. Of course, the increased thickness increases the amount of dynamical scattering, but we believe this does not invalidate the results. The major effects are

increased scattering to higher angles, but this should not eliminate the symmetries measured in the first order diffraction ring.

Representative simulations below of the same area with different thicknesses shows what we have written in the text, mainly that increasing thickness emphasizes lower order symmetries but does not invalidate the analysis. These electron scattering simulations were performed on the same BMG structure used for the MD simulation as in the paper.

Figure 1 NBED simulations of a BMG with two thicknesses, from the MD model in the paper. Column 1 shows no real attenuation of the apparent four-fold symmetry, but column 2 shows the reduction in four/six fold symmetry (dashed red circles) as thickness was increased.

Figure 2 Additional simulations with thicker simulations, showing the same trend of maintaining 2 and 4 fold symmetry throughout a range of thicknesses..

In page 5, how do you determine the spatial resolution of 2nm, when the size of the electron beam is 1.47nm FWHM?

This is the step size of the probe in real space. We have now clarified this in the text.

More details of the in situ loading is provided (10nm then pause), but what about the drift rate? 70 sec per frame is very long. Any effort to correct the effect of drift?

We are happy to provide more clarification on the loading procedures used during this experiment. Luckily, the milled hole in the center helps as a rather large fiducial for registration during experimentation, and the lack of any apparent drift artifacts with its edges, or the outside of the sample provide evidence that spatial drift is negligible. This is now mentioned in the text.

The reviewer also might be worried about the drift of the nano-load frame that we use for the deformation, although this is not clear. As the test was done under displacement control (Supplementary Figure 2), there is some linear load drift. If this was significant, there would be a linear change in strain from top to bottom of the sample (the slow scan direction), but this is not seen.

There is still no mention about how the strain was determined. I know this was explained in Ref 41, but it will be necessary to at least have them included in the methodology section in this paper, as it is important. Reading this paper only gives no idea of how they are determined. It is also puzzling because, if the patterns are like the ones shown in Fig.1, then I think they look quite different from the ones shown in Ref 41 (which is possibly due to the difference in the probe size?)

The method for measuring strain was taken directly from Ref. 41. This is now included in the methods section. Comparing how the diffraction patterns look, in ref. 41 only the mean diffraction patterns are shown from each NBED stack, so it is natural that they look smoother and contain less structure. We have provided here the mean diffraction pattern from ref. 41 and our dataset for comparison.

Figure 3 a) ref. 41 mean diffraction pattern. b) Current work's mean diffraction pattern.

Reviewers' comments:

Reviewer #4 (Remarks to the Author):

Pekin et al. have used nanobeam electron diffraction (NBED) on a bulk metallic glass (BMG) sample undergoing in situ tensile straining to investigate local changes in the symmetry of structural units under plastic strain, before shear banding and failure. They report reduced 2- and 4-fold symmetry in diffraction in regions of the sample that exhibit the largest localized strain. They compare the experiments to MD simulations with a similar geometry which show reduced icosahedral order in regions of high strain. They conclude that the reduced local symmetry in experiments confirms the shear transformation zone (STZ) hypothesis for nucleation of shear bands in MGs.

From my reading of the manuscript and the previous round of reviews, there are three key issues with the manuscript that must be considered:

1. Are the experimental results valid? In particular: (a) Are the authors cherry-picking the subset of the data which shows the largest reduction in symmetry? (b) Is the TEM sample too thick for the experiments to be meaningful?
2. Is the comparison to simulations correctly made and meaningful? In particular, what is the connection between the icosahedral order calculated from the MD structures and the 2- and 4-fold symmetry observed in the experiments?
3. What is the strongest conclusion regarding the existence (or not!) of STZs on the basis of the experiments? (STZs in simulations being generally well-established.)

Let us consider each question in turn.

1a, data subset selection: I am less concerned about this aspect of the work than the previous Reviewer 2. I find the statistics of Supplementary Figure 1 and 4, in addition to the data presented in the manuscript to be reasonably convincing. They show that the conclusions are relatively robust against bias in the selection of the data subsets.

That said, I believe the manuscript would be significantly strengthened by inclusion of a graph of rotational symmetry coefficient vs strain for every pattern in the entire field of view for the 30 nm displacement data and perhaps one additional data set (e.g. 0 nm). Such a graph would test, directly and without selecting a subset of the data, the core contention of the manuscript that angular symmetry in diffraction is reduced by strain. The graph would undoubtedly be noisy, given the high background of structural disorder and the small magnitude of the signals, but the correlation should be visible.

1b, sample thickness: This issue is more troubling. As Im et al. (Ref 52, Fig 2c,d) showed on very similar samples, samples that are 20 nm thick show clearly higher Fourier coefficients in the Friedel-allowed even symmetries than in the Friedel-forbidden odd symmetries. Samples that are 42 nm thick show equal Fourier coefficients for even and odd symmetries. These results suggest strongly that 42 nm is too thick to obtain reliable quantitative measures of rotational symmetry for individual clusters. Why this should be is not explained in detail, but it is likely to be either plural elastic scattering or chance overlap of multiple diffracting clusters through the thickness.

The samples used in this manuscript are 80-90 nm thick, suggesting that it will be difficult to measure intrinsic rotational symmetries of single clusters using NBED. However, I don't think this obviates the conclusion of reduced symmetry caused by strain. Instead, as the author suggest on p. 8-9 "Additionally, it should be noted that since this measurement is projected through the sample thickness, if four-fold symmetry arises from the overlap of two two-fold symmetry elements, this arrangement of atoms is more easily displaced as to reduce diffraction during deformation than a single cluster, further contributing to an increased likelihood of reduction of

four-fold order when compared to two-fold order." This point could be emphasized and claims about specific reduction in angular symmetry softened, while the primary conclusion of the manuscript is preserved.

If the authors wish to argue that the detailed symmetries are correct despite the sample thickness, additional evidence will be required. They present some simulations in Fig 2 of the rebuttal, but the nature of the simulations and the analysis is not clear just from the array of pictures. To be clear: detailed discussion of TEM sample thickness is a sidelight to the main conclusion of the article, which I do not view as required unless the authors wish to make specific strong claims about the nature of the symmetry that is reduced by strain.

2, comparison to simulations: This is the weakest part of the paper. The findings of the MD alone are neither novel nor particularly detailed, and the connection to experiments is somewhere between weak and misleading. An incautious reader might come away with the impression that the authors are claiming experimental observation of reduced icosahedral order associated with shear band nucleation in MGs. That claim is very far from being justified, since the experiments by themselves say nothing about icosahedral or other forms of order.

What, then, to do with this aspect of the work? One path would be to calculate a different descriptor from the MD structures which is more closely related to the experiments. Perhaps a local structural symmetry parameter like the bond-order parameter would be helpful, perhaps after incorporating the inherent 2D projection of electron imaging. Or perhaps systematic nanodiffraction simulations from the MD structures, analyzed following the experiments would be helpful. The key claim is reduced rotational symmetry in the structure with strain, which is hinted at but not directly shown by reduced icosahedral order.

Another path would be to try to revise the text with the current results to foreground the lack of experimental evidence for icosahedral order in experiments, and the use of icosahedral Voronoi polyhedra as a proxy for rotational symmetry. This would require careful attention in the text to what can and can't be supported by the experiments.

3, STZs: I believe that the strongest claim the manuscript can make that is well-supported by the experimental evidence is that this BMGs exhibits a distinct structural change, at the nanometer scale, caused by strain. That by itself is an exciting and novel conclusion! The data further suggest that the nature of this structural change is a reduction in local rotational symmetry, although detailed analysis is rendered questionable by the thickness of the sample.

It is tempting (and interesting!) to identify the observed structural changes with activation of STZs since they occur one deformation step before the sample fractures due to formation of a shear band. STZ models require a localized structural change before and after activation, as observed. However, they also often feature concepts of "incipient" STZs – local regions more likely to deform at a given stress – or avalanche activation, in which an activated STZ increases the likelihood of STZ activation in its immediate environment, neither of which can be shown from the current data.

As a result, I recommend that the authors revise to report that current observations "as consistent with" or "support" the STZ model of deformation, rather than "confirm" it, and provide "evidence for" rather than "evidence of" STZs nucleating shear bands. These may seem like minor tweaks, but I think that precision in acknowledging the limitations of the current experiments will improve the impact and longevity of the manuscript.

Finally, should this manuscript be published in Nature Communications? I recommend that yes, it

should be. The basic experimental finding of reduced rotational symmetry in diffraction created by strain in a metallic glass is novel and potentially far-reaching. To date, such a connection has only been possible in simulations which occur at such small samples sizes and high strain rates that there is always a question about whether they apply to real materials. This work will find substantial interest in the community of researchers working on metallic glass mechanical properties and the mechanical properties of disordered materials more broadly, and I expect it will stimulate significant further experimental and simulation work in the future.

Reviewers' comments:

Reviewer #4 (Remarks to the Author):

Pekin et al. have used nanobeam electron diffraction (NBED) on a bulk metallic glass (BMG) sample undergoing in situ tensile straining to investigate local changes in the symmetry of structural units under plastic strain, before shear banding and failure. They report reduced 2- and 4-fold symmetry in diffraction in regions of the sample that exhibit the largest localized strain. They compare the experiments to MD simulations with a similar geometry which show reduced icosahedral order in regions of high strain. They conclude that the reduced local symmetry in experiments confirms the shear transformation zone (STZ) hypothesis for nucleation of shear bands in MGs.

From my reading of the manuscript and the previous round of reviews, there are three key issues with the manuscript that must be considered:

1. Are the experimental results valid? In particular: (a) Are the authors cherry-picking the subset of the data which shows the largest reduction in symmetry? (b) Is the TEM sample too thick for the experiments to be meaningful?
2. Is the comparison to simulations correctly made and meaningful? In particular, what is the connection between the icosahedral order calculated from the MD structures and the 2- and 4-fold symmetry observed in the experiments?
3. What is the strongest conclusion regarding the existence (or not!) of STZs on the basis of the experiments? (STZs in simulations being generally well-established.)

Let us consider each question in turn.

1a, data subset selection: I am less concerned about this aspect of the work than the previous Reviewer 2. I find the statistics of Supplementary Figure 1 and 4, in addition to the data presented in the manuscript to be reasonably convincing. They show that the conclusions are relatively robust against bias in the selection of the data subsets.

That said, I believe the manuscript would be significantly strengthened by inclusion of a graph of rotational symmetry coefficient vs strain for every pattern in the entire field of view for the 30 nm displacement data and perhaps one additional data set (e.g. 0 nm). Such a graph would test, directly and without selecting a subset of the data, the core contention of the manuscript that angular symmetry in diffraction is reduced by strain. The graph would undoubtedly be noisy, given the high background of structural disorder and the small magnitude of the signals, but the correlation should be visible.

We thank the reviewer for the excellent suggestion. This is now shown in Supplementary Figure 2 (reproduced below, without caption). As we now show, in both the two and four fold symmetry coefficients, under deformation, there is a (slight) negative correlation between order and strain over the whole field of view.

The text describing this figure now reads “Furthermore, Supplementary Fig. 2 plots the NFCs versus strain over the whole field of view ($> 22,000$ probe positions) before and during deformation. While there is no correlation between strain state and amount of order in the unstrained state, there exists a small but measurable negative correlation between strain and order under deformation.”

1b, sample thickness: This issue is more troubling. As Im et al. (Ref 52, Fig 2c,d) showed on very similar samples, samples that are 20 nm thick show clearly higher Fourier coefficients in the Friedel-allowed even symmetries than in the Friedel-forbidden odd symmetries. Samples that are 42 nm thick show equal Fourier coefficients for even and odd symmetries. These results suggest strongly that 42 nm is too thick to obtain reliable quantitative measures of rotational symmetry for individual clusters. Why this should be is not explained in detail, but it is likely to be either plural elastic scattering or chance overlap of multiple diffracting clusters through the thickness.

The samples used in this manuscript are 80-90 nm thick, suggesting that it will be difficult to measure intrinsic rotational symmetries of single clusters using NBED. However, I don't think this obviates the conclusion of reduced symmetry caused by

strain. Instead, as the author suggest on p. 8-9 “Additionally, it should be noted that since this measurement is projected through the sample thickness, if four-fold symmetry arises from the overlap of two two-fold symmetry elements, this arrangement of atoms is more easily displaced as to reduce diffraction during deformation than a single cluster, further contributing to an increased likelihood of reduction of four-fold order when compared to two-fold order.” This point could be emphasized and claims about specific reduction in angular symmetry softened, while the primary conclusion of the manuscript is preserved.

If the authors wish to argue that the detailed symmetries are correct despite the sample thickness, additional evidence will be required. They present some simulations in Fig 2 of the rebuttal, but the nature of the simulations and the analysis is not clear just from the array of pictures. To be clear: detailed discussion of TEM sample thickness is a sidelight to the main conclusion of the article, which I do not view as required unless the authors wish to make specific strong claims about the nature of the symmetry that is reduced by strain.

We are in agreement in that we believe the discussion of sample thickness is a sidelight to the main conclusion. We have emphasized in the paper the inherent difficulties of mapping the exact origin of symmetry in our sample to individual clusters of atoms, stating that “It must be emphasized that, due to plural scattering and projection effects inherent to this experiment, decrease in any symmetry order coefficient (while indicative of structural change) is difficult to directly map to equivalent reductions of symmetry in individual singular clusters of atoms.”

2, comparison to simulations: This is the weakest part of the paper. The findings of the MD alone are neither novel nor particularly detailed, and the connection to experiments is somewhere between weak and misleading. An incautious reader might come away with the impression that the authors are claiming experimental observation of reduced icosahedral order associated with shear band nucleation in MGs. That claim is very far from being justified, since the experiments by themselves say nothing about icosahedral or other forms of order.

What, then, to do with this aspect of the work? One path would be to calculate a different descriptor from the MD structures which is more closely related to the experiments. Perhaps a local structural symmetry parameter like the bond-order parameter would be helpful, perhaps after incorporating the inherent 2D projection of electron imaging. Or perhaps systematic nanodiffraction simulations from the MD structures, analyzed following the experiments would be helpful. The key claim is reduced rotational symmetry in the structure with strain, which is hinted at but not directly shown by reduced icosahedral order.

Another path would be to try to revise the text with the current results to foreground the lack of experimental evidence for icosahedral order in experiments, and the use of icosahedral Voronoi polyhedra as a proxy for rotational symmetry. This would require

careful attention in the text to what can and can't be supported by the experiments.

We have taken care previously to emphasize that in the simulations icosahedral order is only a proxy for local structural order seen in the experiment, for example, writing, “while in this experiment it is not possible to directly link the structural transition mechanism under deformation to icosahedral order, we do see for the first time structural transition under deformation in the in situ NBED experiment” (p. 10). Among these caveats we have already provided, we have also added that local icosahedra in the MD simulation are a proxy for rotational symmetry, as suggested above. This is not a bad proxy given the number of rotation centers icosahedra have, and their predominance in the BMG literature as a common structural unit. We hope the reviewer will find this change acceptable.

3, STZs: I believe that the strongest claim the manuscript can make that is well-supported by the experimental evidence is that this BMGs exhibits a distinct structural change, at the nanometer scale, caused by strain. That by itself is an exciting and novel conclusion! The data further suggest that the nature of this structural change is a reduction in local rotational symmetry, although detailed analysis is rendered questionable by the thickness of the sample.

It is tempting (and interesting!) to identify the observed structural changes with activation of STZs since they occur one deformation step before the sample fractures due to formation of a shear band. STZ models require a localized structural change before and after activation, as observed. However, they also often feature concepts of “incipient” STZs – local regions more likely to deform at a given stress – or avalanche activation, in which an activated STZ increases the likelihood of STZ activation in its immediate environment, neither of which can be shown from the current data.

As a result, I recommend that the authors revise to report that current observations “as consistent with” or “support” the STZ model of deformation, rather than “confirm” it, and provide “evidence for” rather than “evidence of” STZs nucleating shear bands. These may seem like minor tweaks, but I think that precision in acknowledging the limitations of the current experiments will improve the impact and longevity of the manuscript.

We thank the reviewer for the suggestion, and agree with their characterization. We have softened our claims on the STZ model of deformation being confirmed to supported in the conclusions, and also changed the wording to “evidence for” in the abstract.

Finally, should this manuscript be published in Nature Communications? I recommend that yes, it should be. The basic experimental finding of reduced rotational symmetry in diffraction created by strain in a metallic glass is novel and potentially far-reaching. To date, such a connection has only been possible in simulations which occur at such small samples sizes and high strain rates that there is always a question about whether they apply to real materials. This work will find substantial interest in the community of researchers working on metallic glass mechanical properties and the mechanical

properties of disordered materials more broadly, and I expect it will stimulate significant further experimental and simulation work in the future.

We thank the reviewer for the support, and look forward to continuing research based on this work.

REVIEWERS' COMMENTS:

Reviewer #4 (Remarks to the Author):

The authors have answered my criticisms from the previous draft. I believe the manuscript is a balanced, rigorous presentation of some important experimental results for the field of metallic glasses. Connecting strain to atomic structure is a significant advance in the field. I recommend this manuscript for publication in Nature Communications.